# The Regulatory Role of Key Metabolites in the Control of Cell Signaling

**DOI:** 10.3390/biom10060862

**Published:** 2020-06-05

**Authors:** Riccardo Milanesi, Paola Coccetti, Farida Tripodi

**Affiliations:** Department of Biotechnology and Biosciences, University of Milano-Bicocca, 20126 Milan, Italy; r.milanesi2@campus.unimib.it (R.M.); farida.tripodi1@unimib.it (F.T.)

**Keywords:** Snf1/AMPK/SnRK1, Ras/PKA, TORC1, RTG, calcium, glucose, glycolysis, TCA, amino acids, protein-metabolite interaction

## Abstract

Robust biological systems are able to adapt to internal and environmental perturbations. This is ensured by a thick crosstalk between metabolism and signal transduction pathways, through which cell cycle progression, cell metabolism and growth are coordinated. Although several reports describe the control of cell signaling on metabolism (mainly through transcriptional regulation and post-translational modifications), much fewer information is available on the role of metabolism in the regulation of signal transduction. Protein-metabolite interactions (PMIs) result in the modification of the protein activity due to a conformational change associated with the binding of a small molecule. An increasing amount of evidences highlight the role of metabolites of the central metabolism in the control of the activity of key signaling proteins in different eukaryotic systems. Here we review the known PMIs between primary metabolites and proteins, through which metabolism affects signal transduction pathways controlled by the conserved kinases Snf1/AMPK, Ras/PKA and TORC1. Interestingly, PMIs influence also the mitochondrial retrograde response (RTG) and calcium signaling, clearly demonstrating that the range of this phenomenon is not limited to signaling pathways related to metabolism.

## 1. Introduction

The robustness of a biological system is defined as the ability to maintain its principal functionalities regardless of external and internal variations [1]. The concept of robustness could be applied at systems of different order of magnitude. This property generally arises from a specific architecture of the systems itself, that may be a cell, a metabolic pathway, a microbial community or a multicellular organism [2]. Robust systems are typically organized with a bow-tie architecture, where a lot of input signals converge on a limited number of elements, from which a large number of outputs emerge [3]. Metabolic and signaling networks show such an organization, with several metabolites and signals that converge on the central metabolism or on a limited number of signaling pathways [3].

A key aspect of every robust system is the ability to monitor the environmental conditions and adapt its functions to perturbations [4]. For example, from a system biology point of view, it is common to see the maximization of growth as the objective function of a cell. To do so, metabolism, growth and cell cycle progression must be coordinated and finely tuned to fully exploit the available nutrients [5,6]. This requires a precise crosstalk between metabolism, signal transduction pathways and cell cycle progression. Therefore, some molecular events should connect given metabolic reactions with signal transduction cascades, in order to coherently integrate a given flux distribution with the activity of regulatory proteins, such as the kinases upstream to signaling pathways. However, recent publications support the idea that we still need more data to understand the role of specific metabolites in controlling proteins activity at a systems level [7,8].

Since the beginning of the omics era, emergent technologies have focused on the qualitative and quantitative analysis of chemically homogeneous groups of molecules, giving rise to transcriptomics, proteomics and metabolomics. Nevertheless, the interpretation of the data acquired with these approaches is not always straightforward, since the functioning of a biological system emerges by the interactions between chemically different molecules.

The molecular interaction between proteins and DNA has been systematically investigated for many years. On the contrary, the systematic study of the interaction between proteins and metabolites (PMI, protein-metabolite interaction) has started to be explored only very recently. Although recent reports suggest an important role of PMIs in the regulation of signaling [9,10], most publications still focus on the effect of the metabolites belonging to glycolysis and TCA cycle on the activity of the enzymes of the same pathways [7,8,11]. Therefore, a systematic analysis of the interaction between molecules of the central carbon metabolism and the proteins that compose the signal transduction network is still largely missing.

Here we review the known PMIs taking place between signal transduction pathways component and metabolites belonging to the central metabolism, focusing on the output that these interactions have on the activity of the key cellular pathways.

## 2. The Biochemistry of the Protein-Metabolite Interaction (PMI)

From a biochemical point of view, the final result of a Protein-metabolite interaction is a variation in the protein activity due to a conformational change associated with the binding of a small molecule. NMR and single-molecule spectroscopy studies revealed that proteins can undergo conformational changes both when bound and unbound to a small molecule, as a consequence of thermally activated exchanges between the higher and the lowest energy conformations [12,13]. This means that folded proteins in their basal conformation can sample the energy landscape moving towards relatively higher energy states, that are the one proposed to bind the ligands [14,15]. Because of this, an allosteric regulation due to the binding of a small molecule on its target can be explained by two different models: conformational selection and induced fit [16,17].

According to the conformational selection model, a protein complex spontaneously undergoes a conformational change moving from a basal state to an excited one, characterized by a higher energy and, for this reason, less abundant [17]. In this mechanism, the ligand binding takes place after the conformational change and results in the stabilization of the excited conformation, therefore leading to the enrichment of the bound molecular species [18,19]. On the contrary, in the induced fit model, ligand binding takes place with the basal conformation of the target before any conformational change, this results in an intermediate bound complex still in the initial conformation and characterized by a higher energy. The relaxation of this structure is achieved through the fitting of the target to the bound molecule, as recalled by the definition of this mechanism [16,20].

From a technological point of view, a deep understanding of the control mechanism of a given protein may be an important insight in order to rationally design drugs to target it. Alongside classical drugs that target the active site of a protein, nowadays research is focusing also on the development of molecules that may act as allosteric effectors [21,22]. For example, a possible way to exploit conformational selection is to develop molecules that bind to a protein in a high energy conformation and stabilize it, shifting the equilibrium away from the active conformation [21,23].

To achieve this aim, the detailed biochemistry of the regulation of the target must be understood, also considering possible allosteric interactions and conformational changes. 

## 3. The Main Pathways of Nutrient Sensing and Cellular Signaling

Cell growth and progression through the cell cycle must be coordinated, as already reported, with the availability of nutrients [6,24,25]. In eukaryotic organisms, such as yeasts and mammals, but also in plants, the most important nutrients are carbon and nitrogen sources, whose availability represents a positive signal for cellular proliferation. The perception of these nutrients relies mainly on three different signaling pathways, centered on three widely conserved protein kinases: PKA, AMPK and TORC1 complexes [26,27]. The activation of such pathways takes place through a molecular interaction between nutrients (more often a derived molecule) and a receptor. This gives rise to a transcriptional and post-translational response, that results in the regulation of genes and proteins required for protein translation, cell cycle progression and response to nutritional stress [28,29].

Mitochondrial dysfunction is signaled to the nucleus through the retrograde response pathway (RTG), which controls the expression of genes required for respiration and consumption of alternative carbon sources [30]. In addition, the release of cytoplasmic calcium from stores, which allows a huge variety of functions, from muscular contraction to neuronal activities in multicellular organisms, is a highly controlled mechanism.

A relevant part of the crosstalk between signal transduction and metabolism is known to take place at the plasma membrane, where nutrients perception elicits signal transduction resulting in the modification of gene expression and in the post-translational modification of key metabolic enzymes [31]. Some examples are the perception of glucose by Snf3 and Rgt2 in *Saccharomyces cerevisiae* and the perception of amino acids, ammonium, sulphate and phosphate through transceptors in the same organism (see [5,32] for a detailed description of these phenomena). Despite this, the aim of this review is to highlight the mechanism by which intracellular primary metabolites affect the activity of signaling pathways in different eukaryotic species, going beyond the classic description of signaling control on metabolism. All the pathways cited before are controlled by protein-metabolite interactions and will be extensively discussed in the present review. Interestingly, the involvement of PMIs in the control of mitochondrial retrograde response (RTG) and calcium signaling exemplify that this mechanism is not limited only to metabolism related pathways [30], but can be considered a general feature in cell signaling.

### 3.1. Protein-Metabolite Interactions Controlling the Ras/PKA Pathway

Glucose is the preferred carbon source for many organisms, including yeasts, but also for many healthy and unhealthy mammalian cell types. In the budding yeast *Saccharomyces cerevisiae*, glucose availability activates the cAMP mediated Ras/PKA pathway through extracellular and intracellular mechanisms [33]. The extracellular system relies on the G-protein coupled receptor Gpr1, which, once bound to glucose, interacts with Gpa2, the α-subunit of a heterotrimeric G-protein complex. This promotes Gpa1 loading with GTP and its release from the complex [34]. In parallel, intracellular perception of glucose is mediated by the monomeric GTP-binding protein Ras, encoded by the *RAS1* and *RAS2* genes that, in the presence of glucose, exchanges GDP with GTP, in a reaction catalyzed by the GEF (Guanine Nucleotide Exchange Factor) Cdc25 [35,36].

The two different mechanisms for Ras/PKA activation converge on the adenylate cyclase (AC) Cyr1, whose catalytic activity is triggered by the interaction with Ras2 and Gpa2, resulting in a strong increase of intracellular cAMP concentration. Interestingly, the Gpr1/Gpa2 pathway requires Ras pre-activation of Cyr1, with the result that the signaling of the availability of extracellular glucose is subordinated to its actual consumption into glycolysis [37]. Glucose-mediated increase in cAMP concentration results in PKA activation. This is achieved through the binding of the second messenger cAMP to Bcy1, the inhibiting subunit of PKA, inducing its dissociation from the catalytic subunits (Tpk1,2,3) [38]. Active PKA promotes fermentative growth by stimulating glycolytic flux. Accordingly, it represses processes associated with slow growth, such as mitochondrial respiration, stress responses and carbohydrate storage [5,31].

The Ras/PKA pathway is strongly influenced by the interactions between its components and metabolites, that coordinate the activation state of the kinase with the metabolic state of the cell. In fact, as shown in Figure 1, cAMP synthesis is responsive to glucose transport into the cell and its phosphorylation to glucose-6-phosphate (G6P), while it has been discovered only recently that fructose-1,6-biphosphate (F1,6bP) is the glycolytic metabolite that actually controls Ras activity (Figure 1) [9,39].

Peeters and coworker started from the observation that yeast hexokinases (Hxk1 and Hxk2) are negatively controlled by the synthesis of trehalose, catalyzed by Tps1 starting from G6P [40]. *TPS1* deleted cells show a higher glycolytic flux and a stronger increase in Ras activation, with a higher amount of Ras-GTP than the wild type. This phenotype is reverted in cells completely impaired in the production of F1,6bP, due to the deletion of both phosphofructokinase genes, *PFK1* and *PFK2* [41]. In addition, Ras activation positively correlates with F1,6bP concentration in permeabilized spheroplasts and remarkably, F1,6bP binds to the complex formed by the catalytic domain of hSOS (the human homolog of Cdc25) and H-RAS, one of the human isoforms of Ras (Figure 1) [9]. This interaction has been proposed to take place at the interface of the complex formed by Cdc25 and Ras; indeed, point mutations of Cdc25 residues involved in the binding impair F1,6bP activation of Ras [9]. These results strongly suggest that the glycolytic flux is connected to the PKA pathway by a molecular interaction between F1,6bP and the Ras/Cdc25 complex, which results in the activation of the G-protein and in the increase of cAMP concentration [9].

Furthermore, this allosteric regulation is coherent with the proposed role of F1,6bP as glycolytic flux-sensor. In fact, F1,6bP concentration positively correlates with sugar uptake rate and is associated with yeast fermentative metabolism [42]. *S. cerevisiae* supports its growth on glucose with a fermentative metabolism unless the concentration of this carbon source is very low. Since the Ras/PKA pathway responds to glucose availability and supports fermentation, its regulation by F1,6bP looks perfectly coherent with its function and gives mechanistic insights of how F1,6bP level could correlates with glycolytic flux and ethanol production.

Remarkably, as shown in Figure 1, F1,6bP have been clearly demonstrated to bind with hSos1, the mammalian Cdc25 homologue, suggesting that F1,6bP control on the Ras/PKA pathway is also conserved in higher eukaryotes [9].

As described above, cAMP production by adenylate cyclase (AC) plays a central role in the activation of PKA. Prokaryotes and vertebrates present two classes of AC, distinguished in transmembrane (tmAC) and soluble (sAC) adenylate cyclases (as indicated also in Figure 1) [43]. tmACs represent the “classical” adenylate cyclases responsive to G-protein coupled receptors, while sACs result in being activated by bicarbonate ions, inducing a strong increase in their catalytic activity [44,45]. Crystallographic analysis of human and of cyanobacteria *Spirulina platensis* sACs revealed similar structures and a conserved interaction with bicarbonate ions [46,47]. *S. platensis* sAC presents two different ATP binding sites, while a bicarbonate specific binding site is still undiscovered due to the difficulties in sAC co-crystallization with this ion [46]. On the contrary, in human sAC one of the putative ATP binding sites is reported to bind with bicarbonate ions too [47]. Despite this difference, bicarbonate interaction induces a conformational change in both these enzymes. For *S. platensis* sAC this leads to the onset of catalysis, due to the closure of the active site and to the exit of ATP pyrophosphate group thanks to its re-orientation towards Arg1117 [46].

In mammals, sAC is prevalently expressed in testis. Bicarbonate stimulation of sAC has been reported to be pivotal in the activation of processes involved in fertility and, indeed, its depletion results in sterile mice [48]. In addition, sAC is also expressed in other organs, such as kidney, and in the choroid plexus, and its function has recently been linked to the oncogenic Ras-dependent activation of pro-tumoral macropinocytosis [44,49]. This process has been reported to support cancer growth, gaining amino acids from the degradation of extracellular proteins [50]. Interestingly, the onset of this process does not require the activity of the tmAC, but that of the sAC, which is stimulated by the bicarbonate imported from the extracellular matrix by SLC4A7 transporter. In such a way, the metabolism of cancer cells expressing an oncogenic Ras results to be connected with the extracellular acidification, coupling pH homeostasis and metabolic adaptation of mutant Ras-driven tumors (as schematically shown in the right part of Figure 1) [49].

Phylogenetic studies revealed the existence of four different classes of AC and eukaryotes have been reported to express only class III AC. Both transmembrane and soluble mammalian AC belong to this class but, strikingly, no soluble AC have been found in *S. cerevisiae*, *Caenorhabditis elegans*, *Drosophila melanogaster* and plants [51]. Despite this, classical ACs from *Candida albicans* and *Cryptococcus neoformans* are activated by bicarbonate, promoting virulence associated processes such as filamentous growth and encapsulation (see the left part of Figure 1) [52,53]. This highlights that given the differences between the ACs expressed by mammals and yeasts, such enzymes can be valuable therapeutic targets for the treatment of *Candida* and *C. neoformans* infections.

The effect of bicarbonate on AC is also physiologically relevant for sporulation of diploid cells of *S. cerevisiae.* During this process, the Ras/PKA pathway needs to be finely tuned, since low PKA activity is required to trigger meiosis, and cAMP level oscillates during it [54]. An environmental condition that triggers sporulation is the availability of acetate as sole carbon source, while its concentration positively correlates with sporulation efficiency through a mechanism that require Ras/PKA. In particular, as schematized in Figure 1, acetate-dependent sporulation is triggered by bicarbonate ions produced by acetate oxidation in the TCA cycle and CO_2_ hydration by the carbonic anhydrase (Figure 1) [55]. Bicarbonate interacts with the adenylate cyclase Cyr1, and the mutation of one of the lysine involved in the interaction (Cyr1K1712A) results in the loss of the modulation of the sporulation efficiency by acetate concentrations. The same phenotype is observed in Ras and PKA depleted cells and demonstrates that acetate modulation of sporulation passes through the Ras/PKA pathway and the modulation of Cyr1 activity, due to acetate oxidation and bicarbonate production [54].

All these results suggest that despite yeasts not expressing soluble adenylate cyclase, bicarbonate control of AC activity is conserved in bacteria, fungi and mammals, with fungi canonical ACs being controlled both by G-proteins and bicarbonate ions, as indicated in Figure 1.

### 3.2. Protein-Metabolite Interactions Controlling SNF1/AMPK/SnRK1 Pathway

As described above, glucose represents a positive signal for cell growth through the Ras/PKA pathway. Its presence is associated with a good nutritional condition which ensures enough carbon and energy to grow and proliferate [5]. However, since nutrient availability is only one of the parameters that could influence the energy balance, all eukaryotic systems have evolved signaling pathways able to monitor the global energetic status of the cell. These pathways are centered on the protein kinases Snf1, AMPK and SnRK1, depending on the organism (yeast, mammals or plants), as better explained below [56,57,58].

These kinases operate as heterotrimeric complexes composed of a catalytic subunit α and two regulatory subunits, β and γ. SNF1/AMPK/SnRK1 complexes are active in low energy conditions and are tightly connected to the cellular metabolism. The key event in their regulation is the phosphorylation of the catalytic subunit at the level of a functionally conserved threonine (T210 in yeast, T172 in humans and T175 in plants) which activates these kinases and, as a consequence, energy-expensive processes are switch off [56,57,58].

In *S. cerevisiae*, Snf1 protein kinase was discovered in a genetic screening of mutants unable to grow on non-fermentable carbon sources [59]. Because of this, the SNF1 complex has generally been recognized as being active in low glucose conditions or in the presence of alternative carbon sources such as ethanol, glycerol and galactose. In these conditions, SNF1 activity is pivotal to rewire the yeast metabolism and to sustain cell growth with the induction of fatty acid oxidation, mitochondrial respiration and gluconeogenesis [29]. Despite this, recent reports suggest the involvement of SNF1 in the control of metabolism also under glucose repression [60,61].

While the Snf1 upstream kinases, Sak1, Tos3 and Elm1 are constitutively active (with Sak1 being the primary upstream kinase), the phosphatase activity of Reg1/Glc7 is strongly induced by availability of glucose [62]. Although Reg1-Glc7 PP1 plays the major role, also Ptc1 protein phosphatase 2C and Sit4 type 2A-related phosphatase were shown to contribute to glucose-regulated dephosphorylation of SNF1 [63,64].

A decrease in the ATP-to-ADP intracellular ratio is representative of a perturbation of the energy homeostasis and SNF1 activation state is controlled by a protein-metabolite interaction with ADP. As presented in Figure 2, the result of this interaction is the protection of the regulatory loop from phosphatase activity, with a resulting higher phosphorylation level of the threonine 210 [65]. Despite this clear evidence, the ADP-dependent protection of Snf1 phosphorylation is still controversial. At first, the protective effect was attributed to ADP binding with the Bateman (CBS) domains of the γ regulatory subunit [65], but later, an alternative model was proposed, according to which the direct binding of ADP in the catalytic site of the α subunit could promote phosphatase resistance [66]. 

In addition, some data have shown that SNF1 activity negatively correlates with glucose availability and glucose-6-phosphate synthesis, although the molecular mechanism through which glucose represses SNF1 is not fully understood. It has been reported that glucose strongly increases PP1 and PP2A phosphatases activity in a PKA-dependent manner, while Sak1, the major Snf1 upstream kinase, is phosphorylated by PKA (Figure 2) [62,67]. Moreover, SNF1 complex has been reported to influence PKA pathway by phosphorylating Cyr1 [68]. These findings suggest that Snf1 phosphorylation may be linked to glycolysis through the Ras/PKA pathway and that these kinases may be connected by a feed-back mechanism.

A different hypothesis is based on the observation that Reg1/Glc7 can dephosphorylate its target Mig1 also in low glucose conditions, suggesting that PP1 basal activity may also be sufficient to inactivate Snf1. Therefore, glucose inhibition may take place through a change in the exposure of T210, regardless the activity level of the phosphatase [69].

The SNF1 homolog AMPK plays a well conserved role in mammalian cells and, as for yeast Snf1, its activity is controlled by the phosphorylation on a regulatory Thr, T172, by the upstream kinases LKB1 and CaMKK2 [70,71]. AMPK activation reflects an energetic stress of the cell and coherently induces a downregulation of anabolism and the activation of catabolic processes, such as β-oxidation of fatty acids and mitochondrial respiration [72,73].

AMPK is activated by energy depletion sensed as a decrease in the ATP-to-AMP ratio and thanks to AMP-dependent protection of AMPK complex from phosphatase activity (Figure 2) [74,75]. It was reported that AMP interacts with the γ regulatory subunit of AMPK, similarly to what was shown for ADP on yeast SNF1 complex (Figure 2) [65,76]. The γ regulatory subunit presents four different Bateman (CBS) domains, each characterized by an AMP binding site and it has been shown that the weaker site (named site 3 by the authors) mediates AMP control of AMPK. In particular, AMP binding results in the protection from phosphatases and in the increase of the catalytic activity of the complex [75]. Interestingly, the full-length crystal structure of active AMPK shows a closed conformation in which T172 exposure to phosphatases is reduced. On the contrary, the crystal structure of unphosphorylated and inactive AMPK presents a higher exposure of T172 to the medium [77]. Strikingly, point mutations in the auto-inhibition domain of the catalytic subunit involved in the interaction with the AMP-binding site 3, abolish the AMP-induced protection from phosphatases and the increase of the catalytic activity [77,78].

In addition to this complex mechanism connecting AMPK to energy balance, it has been recently reported a connection between AMPK and the glycolytic metabolite F1,6bP, independently from the ATP-to-AMP ratio [10]. Glucose starvation has been demonstrated to increase the affinity of myristoylated AMPK for the scaffold protein AXIN, that favors AMPK-LKB1 interaction, thus promoting AMPK phosphorylation (Figure 2) [79]. This takes place on the late lysosome membrane, where a protein complex composed of Ragulator (late endosomal adaptor, MAP and mTOR activator 1 and 5) and the V-ATPase alternatively recruits mTOR, in high energy conditions, or AMPK, in low energy conditions, playing an important role in the switch from anabolism to catabolism and vice versa [80]. Remarkably, F1,6bP has been reported to promote the dissociation of the AXIN/LKB1/AMPK complex and, in keeping with that, cells lacking phosphofructokinase show a high phosphorylation of AMPK also in presence of glucose [10]. Moreover, aldolases triple knockdown causes AMPK activation regardless glucose concentration, strongly suggesting that F1,6bP effect on AMPK could be mediated by the interaction with these glycolytic enzymes (Figure 2). Aldolase split F1,6bP into glyceraldehyde-3-phosphate (G3P) and dihydroxyacetone-phosphate (DHAP) and F1,6bP binding to aldolase catalytic site is followed by a conformational change of the protein and the formation of a covalent bound between the substrate and the enzyme [81]. Strikingly, ALDOA-K230A point mutant, which is impaired in the formation of this binding, also prevents F1,6bP inhibition of AMPK and AXIN/LKB1/AMPK complex dissociation [10].

These results indicate that AMPK phosphorylation is directly controlled by glucose availability, which is sensed through changes in F1,6bP concentration, in a mechanism mediated by aldolases. This is the first clear demonstration of a regulation of AMPK activity that takes place before and independently from any variation of the energetic status of the cell, directly connecting its activity to glucose metabolism and glycolytic flux [10].

AMPK activity is also linked to glucose metabolism through the interaction with glycogen. AMPK β subunits present a glycogen binding domain (GBD), also known as carbohydrate binding module (CBM), and its interaction with glycogen have been reported to affect AMPK activity. Single α1-6 branched glycogen molecules have been proved to act as allosteric inhibitors of AMPK, reducing its catalytic activity and preventing its phosphorylation by upstream kinases. Meanwhile, glycogen has no influence on AMPK dephosphorylation by upstream phosphatase PP2Cα and PP1 [82]. Such a control clearly suggests that AMPK, besides the perception of AMP/ATP ratio, can also monitor energy availability through the amount of reserve carbohydrates. 

Interestingly, this feature of AMPK seems not to be conserved in yeast and plants. Snf1 complex β subunits Gal83 and Sip2 present a functional GBD that is involved in the control of Snf1 activity, but not through the binding of glycogen [83]; while in plants, the CBDs of the different isoform of β and βγ hybrid subunits do not show the ability to bind glycogen [84]. 

As concerns plants, energy homeostasis is maintained by the SnRK1 complex, the homolog of SNF1 and AMPK. Energy depleting conditions could affect photosynthesis, mitochondrial respiration or carbon allocation and activate SnRK1, that in turn, downregulates anabolic processes and promotes catabolic ones, in a very well conserved fashion [57]. SnRK1 activity is also controlled by a phosphorylation event regarding the T175 in the activation loop, and the active form of SnRK1 is represented by the canonical heterotrimeric complex composed of an α catalytic subunit and two regulatory subunits, β and γ [85]. 

SnRK1 phosphorylation relies on the activity of the upstream kinases GRIK1 and GRIK2, but also on a strong auto-phosphorylation activity; moreover, overexpression of the SnRK1 α subunit is sufficient to increase the catalytic activity of the complex [86]. These observations suggest that the SnRK1 complex is constitutively active, while it is repressed under energy abundance conditions. This sort of control is the reverse of those of SNF1 and AMPK complexes, but is coherent with the general feature of plant cell signaling to prefer negative regulations rather than activations [87]. 

We have described how yeast SNF1 and mammalian AMPK are controlled at two different levels by two different stimuli, one regarding the general energetic status of the cell, and the other responsive to glucose metabolism. Despite the structural similarities with SNF1 and AMPK, SnRK1 does not respond to changes in AMP or ADP concentration, coherently with the fact that βγ Bateman (CBS) domains present mutations in the residues involved in the binding of adenosyl-nucleotides [84,88]. On the contrary, the effect of glucose on SNF1/AMPK/SnRK1 pathway is well conserved also in plants [89]. 

As depicted in Figure 2, trehalose-6-phosphate (T6P), glucose-6-phosphate (G6P) and glucose-1-phosphate (G1P) can directly bind and inhibit SnRK1 [89,90]. Indeed, the availability of trehalose and sucrose signal a good nutritional state of the plant and therefore the possibility to switch on anabolic processes and growth. T6P has been reported to reduce SnRK1 catalytic activity as a non-competitive inhibitor, without affecting enzyme affinity to ATP. Furthermore, G1P influences SnRK1 enzymatic activity following the same model and, interestingly, the combination of T6P and G1P have a synergistic effect on SnRK1, opening to the hypothesis that these metabolites could bind at two different sites. On the contrary, G6P inhibits ATP binding to the catalytic domain and has a cumulative effect when combined with T6P [90]. Eventually, SnRK1 inhibition by T6P was recently reported to depend on T6P binding to KIN10, the catalytic subunit of SnRK1, affecting its interaction with the upstream kinase GRIK1, therefore reducing SnRK1 phosphorylation and enzymatic activity [91].

According to the presented evidence, we can conclude that energy homeostasis in eukaryotic cells is maintained through protein-metabolite interactions in which key metabolites transduce information to a protein-based control system, whose function is to downregulate anabolic processes and to promote catabolism.

### 3.3. Protein-Metabolite Interactions Controlling TOR Activity in Yeast and Mammalian Cells

Amino acids are among the nutrients that most influence the growth of eukaryotic cells. While glucose ensures energy and carbon availability, amino acids are directly connected to protein and nucleotide synthesis. The TOR pathway is deputed to amino acids perception in all eukaryotes and its activation represents a positive signal for cellular growth; active TOR promotes anabolic processes such as protein, lipid and nucleotide synthesis and represses autophagy [92]. As for carbon, for nitrogen sources there is also a precise hierarchy in the preferred amino acids. In mammalian cells, mTORC1 is preferentially activated by leucine and arginine while glutamine is the main activator of the homolog TORC1 complex in yeast [28,93].

Amino acid sufficiency represents an essential input for TOR activation and its sensing largely relies on RAG-GTPase complexes. Amino acids stimulate the loading with GTP of RagA/B in mammalian cells and of Gtr1 in yeast, as well as the loading with GDP of RagC/D and Gtr2 in the two eukaryotic systems, promoting TOR activation [94,95]. The perception of amino acids requires their transport into the cell and Protein-metabolite interactions between key amino acids and regulator proteins upstream of TOR.

As shown in the left part of Figure 3, in *S*. *cerevisiae*, glutamine and leucine activate TORC1 through Pib2 and Gtr/Ego pathways, respectively [95,96]. *PIB2* genetically interacts with *GTR1* and their double deletion leads to synthetic lethality. Moreover, the suppression of Pib2 expression in a *gtr1∆* background completely abolishes TORC1 activation, suggesting that amino acid sufficiency in yeast is exclusively sensed through these protein complexes [96,97].

*PIB2* deletion is reported to impair glutamine activation of TORC1 (Figure 3). Pib2 co-precipitates with TORC1 under amino acids sufficiency and glutamine is reported to enhance Pib2-TORC1 interaction in vitro in a dose-responsive manner. However, in vitro experiments failed to identify a direct interaction between glutamine and purified Pib2 protein; while glutamine interacts with Pib2 when incubated with cell extracts, suggesting that glutamine binding may be indirect and mediated by another Pib2 interactor [96].

The Ego complex (Ego1, Ego2, Ego3) is known to tether the Grt1/Gtr2 GTPase to the vacuole and amino acids, mainly leucine, stimulate Gtr1 loading with GTP and of Gtr2 with GDP, activating the complex [98]. In yeast, TORC1 constitutively localizes at the vacuole, forming punctate structures under amino acid starvation and diffusing all over the membrane in presence of key amino acids. Active Gtr1/Gtr2 complex strongly interacts with TORC1 and induces its activation [95,99]. As represented in Figure 3, leucine perception by the Gtr/Ego pathway relies on the leucyl-tRNA synthetase (LRS), encoded by *CDC60* [100]. Cdc60 presents both an aminoacyl-tRNA-synthetase activity and a proofreading one, functional to the control of the loading with the proper amino acid [101]. Leucine is the most abundant amino acid in proteins and its tRNA synthetase is the most expressed in yeast cells [102,103]. These observations provide a rationale for TORC1 activation by leucine, despite the fact that it is not the preferred nitrogen source in yeast.

Cdc60 has been reported to interact with Gtr1 in the presence of leucine and treatment with the leucyl-tRNA-synthetase inhibitor 1,3-dihydro-1-hydroxy-2,1-benzoxaborole (DHBB) downregulates TORC1 activity [100]. Since DHBB is known to form an adduct at the proofreading site of the enzyme that trap it in the uncharged state, it is likely that leucine starvation may affect Cdc60-Gtr1 interaction due to an increase in tRNA^Leu^ mischarging and the activation of the proofreading activity [104,105]. Eventually, Cdc60 dissociation from Gtr1 may expose it to the interaction with a yet unknown GAP, causing GTP hydrolysis and the inactivation of the complex, leading to downregulation of TORC1 activity [100].

Pib2 and Gtr1/Gtr2 form distinct complexes with TORC1 and independently modulate its activity. Pib2, as Tor1, localizes in puncta on the vacuole under starvation and diffuses all over the vacuole membrane under amino acids sufficiency. Despite the independent functions of Pib2 and Gtr1, a strain deleted in *GTR1* shows an accumulation of both Tor1 and Pib2 puncta even in presence of amino acids, suggesting that Pib2 localization may be controlled by the Gtr1/2 complex [96]. These observation underline the fact that despite some clear evidence, further studies are required to understand the fine tuning of these two different control mechanisms in *S. cerevisiae*.

In mammalian cells, mTOR activity is controlled by both amino acid sufficiency (in particular by leucine, arginine and methionine, as shown in the right part of Figure 3) and growth factors [106,107]. Under amino acid sufficiency, the Gtr1 ortholog RagA/B is loaded with GTP and dimerizes with the Gtr2 ortholog RagC/D charged with GDP. RAG GTPases constitutively localize on the lysosome membrane and amino acids induce their interaction with the Ego ortholog Ragulator, while their interaction with Raptor recruits mTORC1 at the lysosome [108,109]. Once localized at the lysosome, mTORC1 is activated through the interaction with the GTPase RHEB, which is controlled by growth factors through the AKT pathway [107,110]. Therefore, mTORC1 is activated only when nutritional and environmental conditions are advantageous and cellular growth is both feasible and allowed.

As represented in Figure 3, the function of leucyl-tRNA-synthetase (LRS) in mediating leucine perception is conserved also in mammalian cells [111]. Amino acids and leucine availability induce LRS localization at the lysosome membrane, where it interacts with Raptor and mTORC1 complexes. Both LRS knockdown and point mutation of the leucine binding site result in a reduced activation of mTORC1 upon leucine supplementation, demonstrating that leucine availability is perceived through its binding with LRS. Leucine availability also induces the interaction of LRS with RagD; in particular, the aminoacyl-tRNA-synthetase acts as a GAP on RagD, promoting the hydrolysis of GTP to GDP and inducing the activation of mTORC1 [111].

In keeping with the fact that LRS depletion does not totally impair mTORC1 activation, leucine availability in mammalian cells is perceived also through Sestrin2 (see Figure 3), a monomeric protein sharing common folding with the carboxymucolactone decarboxylase (CMD) protein family. The C-terminal domain of Sestrin2 presents a binding pocket specific for leucine, with residues Glu451 and Arg390 interacting with leucine amine and carboxylic group, respectively. The hydrophobic base of the pocket excludes charged and polar amino acids, while large hydrophobic residues are excluded by steric clash and small one cannot form stable interactions. In addition, a lid structure completely covers leucine within the binding pocket and stabilizes the protein structure [112].

Sestrin2 is known to interact with Gator2, the inhibitor of Gator1, a protein complex acting as GAP of RagA/B. During amino acid starvation, Sestrin2 recruits Gator2 presumably promoting RagA/B GTPase activity and therefore mTORC1 inactivation. On the contrary, during amino acid sufficiency, Gator2 inhibits Gator1, promoting the loading of RagA/B with GTP and mTORC1 activation [113,114]. Leucine is reported to bind to Sestrin2 and affects Sestrin2-Gator2 interaction, while a Sestrin2 mutant impaired for the binding with Gator2 fail to activate mTORC1 in the presence of leucine [111]. Eventually, HEK-293T cells expressing a Sestrin2 mutant with a reduce affinity for leucine show a dose-response activation of mTORC1 shifted towards a higher concentration of amino acids, clearly demonstrating that leucine-dependent activation of mTORC1 requires leucine binding to Sestrin2 [112].

As indicated in Figure 3, arginine is another amino acid known to directly affect mTORC1 activity and, also its perception, is centered on the protein Gator2 [106,115]. Amino acid starvation induces Gator2 interaction with Castor1 and Castor2, two homolog proteins presenting four small molecule binding domains (ACT) each. These proteins can combine into homo- and hetero-complexes, and their dimerization is necessary for their interaction with Gator2. Crystallographic analysis revealed that arginine binds Castor1 at the interface between ACT2 and ACT4, presumably arranging the conformation of a glycine-rich loop and therefore allowing for ACT2-ACT4 interaction [116]. I280A point mutation of Castor1 is reported to impair arginine binding with the protein, resulting in a constitutive interaction with Gator2 and in the inactivation of mTORC1, even in the presence of arginine [115,116]. According to these results, arginine availability activates mTORC1 through the inhibition of Gator1 by Gator2, that may be released from Castor1. Arginine binding is likely to induce a conformational change in Castor1 bringing to an altered exposure of key residues required for its interaction with Gator2 [116].

Interestingly, the ACT domains of Castor1 structurally resemble the allosteric regulatory domain of bacterial aspartate kinase (AK), which is known to be sensitive to leucine. Since AK is absent in metazoan and Castor1 is present only in this lineage, it is likely that arginine perception evolved starting form an already existing module, i.e., prokaryotic aspartate kinases [116]. Following this line of evidence, arginine perception could be relevant also in yeast, through the aspartate kinase Hom3. In fact, Hom3 has been reported to genetically interact with component of the EGO complex and its deletion increases rapamycin sensitivity, suggesting the involvement of Hom3 in arginine perception in yeast [117,118].

In mammalian cells, arginine perception relies also on the lysosomal transmembrane protein SLC38A9 (Figure 3). It has been reported that SLC38A9 interacts with Ragulator complex and with a RagB mutant that mimics a constitutive binding with GDP [119]. Moreover, its depletion affects mTORC1 activation by arginine availability. Since SLC38A9 presents a transport activity specific for amino acids and with a preference for arginine, it may link mTORC1 activity with arginine concentration within the lysosome [119]. Interestingly, a recent report suggests that arginine perception by Castor1 and SLC38A9 occur through independent mechanisms, but a residual mild activation of mTORC1 in cells lacking Castor1 and SLC38A9 suggests the existence of additional controls on the pathway [115].

TOR pathway in mammalian cells has been reported to be influenced also by methionine availability and, as indicated in Figure 3, through its conversation into S-Adenosylmethionine (SAM). [120]. Furthermore, SAM level can be influenced also by folate, betaine and vitamin B_12_, suggesting that also these molecules could indirectly influence mTORC1 activity. Methionine perception passes through a PMI between its activated form, SAM and SAMTOR, a protein presenting a class I Rossmann fold methyltransferase domain able to bind SAM [120]. This mechanism is conserved in *D. melanogaster*, which presents the SAMTOR homolog dSamtor, but not in *S. cerevisiae*, in which methionine availability is perceived through PP2A methylation and not through PMIs [121].

Differently from Sestrin2 and Castor complexes, SAMTOR negatively regulates mTORC1 interacting with Gator1 and KICSTOR, suggesting that Gator1 is inactive when bound to SAMTOR. Hence, the binding of SAM to SAMTOR results in the disruption of the SAMTOR-Gator1-KICSTOR complex, releasing Gator1 and activating mTORC1 [120].

The TOR pathway is quite conserved also in plants, even if its regulation by PMIs is much less clear. The *Arabidopsis thaliana* genome encodes functional homologs of TOR (AtTOR) and two TOR-interactors: LST8 (AtLST8-1 and AtLST8-2) and RAPTOR (AtRAPTOR1A and AtRAPTOR1B). On the contrary, no TORC2 homolog has been identified [122]. It was shown that glucose and sucrose (but not other sugars such as fructose, xylose and galactose) stimulate root growth and meristem activation in a TOR-dependent manner [123]. This glucose activation of the TOR pathway is prevented by 2-deoxy-glucose and by inhibitors of the electron transport chain, indicating that glucose needs to pass through glycolysis and the electron transport chain to give TOR activation, as reported in Figure 3 [123]. In addition, it was recently reported that nicotinamide (vitamin B3) affects ATP level and inhibits the glucose-TOR signaling, regulating the circadian clock, meristem activation and root growth [124]. Although the precise molecular mechanism of glucose activation is still unknown, this last study suggests a link between the energetic status of the cell and TOR activity. This leads to the hypothesis that TOR activation by glucose may be mediated by the kinase SnRK1, which phosphorylates RAPTOR as in mammalian cells, where this phosphorylation has an inhibitory role [125,126]. Another proposed hypothesis is that glucose signals could be required for TORC1 dimerization [122]. In mammalian cells, the glucose/energy sensitive Tel2-Tti1-Tti2 (TTT)-RUVBL1/2 complex (which has orthologous genes in plants) is required for TORC1 dimerization and translocation to the lysosome membrane [127]. The assembly of the TTT-RUVBL1/2 complex and its interaction with TOR requires the activity of RUVBL1/2, which is repressed by respiration inhibition, therefore indirectly linking respiration and TOR activation.

In plants, TOR signaling also responds to sulfur availability. Sulfur deficiency causes a decrease in TOR activity, due to a reduction of glucose, maltose, sucrose and TCA cycle intermediates, thus indicating that sulfur shortage is sensed through the same glucose-TOR signaling described above [128].

Finally, although the major TOR activators in yeast and mammals are amino acids, no amino acid sensors have been discovered in plant genomes. In keeping with that, it was shown that TOR activity cannot be stimulates by supplying of an amino acid cocktail in plants [123]. Nevertheless, it was recently reported that mutants which accumulate branched chain amino acids (BCAA) present a higher TOR activity, which leads to an altered organization of the actin cytoskeleton and endomembranes [129]. This evidence suggests that also in plant cells, amino acids can activate the TOR pathway.

### 3.4. Yeast Mitochondrial Retrograde Response Is Influenced by Protein-Metabolite Interactions

Classically defined as the powerhouse of the cell, it is now clear that mitochondria functions go beyond energy production and that mitochondrial health critically define cell fate [130,131,132]. In mammals, yeast and plants, mitochondrial dysfunctions activate retrograde response pathways (RTGs), eliciting the transcription of nuclear genes with the aim of coping with the perturbation. In yeasts, RTG is also triggered by the shift on non-fermentable carbon sources and leads to the expression of genes involved in respiration and consumption of carbon sources alternative to glucose, but is also involved in multi-drug resistance [133,134,135]. As reported in Figure 4, among the different stimuli known to activate the RTG pathway in yeast there is a drop in ATP concentration, even if its level could also be influenced by non-mitochondrial processes [30]. On the contrary, in mammals and in plants, mitochondrial dysfunctions are signaled to the nucleus mainly by the alteration of mitochondrial membrane potential, followed by a strong increase in cytoplasmic calcium without involving PMIs (see [136,137], for further details).

ATP control of the RTG pathway in yeast is based on a PMI. In cells presenting healthy mitochondria (see the left part of Figure 4), the Mks1-Bmh1/2 complex prevents Rtg3 dephosphorylation and therefore its migration into the nucleus. Mitochondria dysfunctions (right part of Figure 4) induce partial dephosphorylation of the transcription factor Rtg3 and its nuclear localization in complex with Rtg1 [138]. The onset of the signaling is centered around Rtg2 that, under mitochondrial stress, binds Mks1 and promotes Rtg1-Rtg3 translocation into the nucleus [139]. Rtg2 is known to present an N-terminal ATP-binding domain similar to the Hsp70/sugar kinase/actin protein superfamily, whose activity results to be pivotal for the RTG onset [139]. It has been reported that ATP strongly impairs Rtg2 interaction with Mks1 at physiological concentrations, while a non-hydrolysable ATP homologue (adenosine 5′-(β,γ- imido) triphosphate) has no effect on Rgt2-Mks1 interaction [140]. These results suggest that ATP must be hydrolyzed in order to induce Mks1 dissociation from Rgt2; yet, ATP interaction with the ATP-binding domain of Rtg2 has not been proved, leaving open the possibility of an indirect sensing of the ATP availability.

Strikingly, the role of ATP in controlling Mks1 activity is also reported in other yeasts, such as *Kluveromyces lactis* and *Kluveromyces waltii*, suggesting the conservation of this control mechanism among Fungi [140]. In conclusion, despite Snf1 conserved role in monitoring the energetic status of the cell, recent evidence suggest that Snf1 and RTG pathways can independently respond to ATP depletion in yeast [127].

### 3.5. Protein-Metabolite Interactions Controlling Calcium Release

Transient changes in the concentration of intracellular calcium are important regulators of cellular signaling in human cells. The increase of intracellular calcium can be due either to Ca^2+^ entry from the extracellular space through L-type Ca^2+^ channels and to Ca^2+^ release form the endoplasmic/sarcoplasmic reticulum (ER/SR). Two major calcium channels exist on the ER/SR, one dependent on calcium ions, the other controlled by inositol-3-phosphate (IP3). The so-called calcium-induced calcium release (CICR) is mediated by Ryanodine receptors (RyRs), homotetrameric channels, which are normally closed when the cytosolic calcium concentration is low (100–200 nM), while they are open when calcium increases up to low micromolar concentrations [141]. The other pathway of calcium release is activated by IP3 binding to IP3 receptors. Hormones, growth factors and neurotransmitters can stimulate the formation of IP3 through activation of G-protein-coupled receptors (GPCR) or tyrosine kinase associated receptors, which are associated with phospholipase C (PLC). PLC catalyzes the formation of diacylglycerol (DAG) and IP3 from phosphatidyl inositol-4,5-bisphosphate, and IP3 stimulates IP3 receptors, inducing calcium release form the endoplasmic reticulum, characterized by brief calcium transients, that can be repeated to give oscillations [142]. Instead, calcium uptake from the cytosol back into the ER/SR relies on the activity of the SERCA (sarcoendoplasmic reticulum Ca^2+^-ATPase) pump, whose function is essential to guarantee calcium homeostasis. Calcium signaling pathway controls several different processes, among which muscle contraction, neuronal functions, gene expression, metabolism, secretion, fertilization, proliferation and defects in this pathway are associated with a wide range of pathological conditions [142].

Cyclic ADP-ribose (cADPR) is a cyclic metabolite of NAD^+^, synthesized by ADP-ribosyl cyclase, and is the main known regulator of calcium release. It induces calcium release from the SR via RyRs [143,144,145,146], it directly binds to SERCA and stimulates calcium sequestration in the SR [147,148,149] and can also regulate calcium influx from outside, activating the calcium influx channel TRPM2, at the cell surface [150].

In addition, as shown in Figure 5, calcium levels are also controlled by glycolysis. More than 30 years ago, it was shown that in pancreatic islets glucose-6-phosphate increased the ATP-dependent calcium content in the ER [151]. The same effect of G6P on SERCA activity was shown in other tissues [152,153], and G6P import in the ER was suggested to be required for calcium sequestration [152]. In addition, in cardiac myocytes, glycolysis is required for excitation-contraction coupling, regulating SR Ca^2+^ release and Ca^2+^ sequestration by SERCA [154,155]. Interestingly, it was shown that F6P and F1,6BP directly stimulate the open probability of RyRs, while pyruvate and lactate inhibit RyRs activity [154], suggesting that glycolysis modulates calcium release through multiple metabolites.

Calcium regulation in yeast is quite different from that in human cells. The main calcium storage organelle in yeast is the vacuole, a large organelle resembling animal lysosome, which contains more than 90% of calcium of the cell. Calcium is imported from the cytosol by the vacuolar Ca^2+^/ATPase Pmc1 and by the H^+^/Ca^2+^ antiporter Vcx1 [156,157,158], while Ca^2+^ efflux out of the vacuole is mediated by Yvc1 [159]. A minor, exchangeable pool of calcium is located in the ER and Golgi apparatus, where proper calcium levels are maintained by the Ca^2+^/ATPases Spf1 and Pmr1, important for the regulation of protein folding and processing of proteins that transit through the secretory pathway [160]. Yeasts also have a calcium channel on the plasma membrane, composed by Cch1 and Mid1, responsive to intracellular Ca^2+^ storage [158]. Although *S. cerevisiae* does not have an IP3 receptor, it was shown that IP3 generation by Plc1 activation is involved in the calcium increase after glucose addition to starved cells [161].

In yeast, calcium signals are generated during mating, upon environmental stresses (such as osmotic and alkaline stress, ER stress, oxidative stress, heat shock), after glucose addition to starved-cells and during mitosis [156]. Calcium shortage was reported to affect cell cycle, cell growth, metabolism and protein folding [162,163].

Interestingly, as in mammalian cells, glucose metabolites were shown to modulate calcium levels also in *S. cerevisiae*. Glucose or galactose addition to starved cells leads to a transient calcium increase due to the opening of plasma membrane calcium channels and requires hexose transport and phosphorylation [164], but also the Gpr1/Gpa2 complex and the phospholipase C Plc1 [161,165].

Interestingly, cells lacking Pmg2, the major isoform of phosphoglucomutase that catalyzes the interconversion of G1P and G6P, present higher G1P levels when grown in galactose and show higher intracellular calcium levels [166], due to elevated Pmc1-dependent vacuolar Ca^2+^ uptake and reduced ER/Golgi Ca^2+^ accumulation [167]. Strikingly, deletion of the phosphofructokinase Pfk2 in a *pgm2Δ* strain restores the balance between G1P and G6P and restores Ca^2+^ concentration to a level comparable to that of wild type cells [168]. These data suggest that the ratio between G1P and G6P, rather than the level of each single metabolite, regulates intracellular calcium levels. However, the molecular mechanism of this regulation in still unknown, since neither G6P nor G1P are able to directly stimulate Ca^2+^ transport into the vacuole [167]. An indirect mechanism may involve the Snf3 pathway, which was shown to sense G6P levels [169] or the Gpr1/Gpa2 pathway, which has already been implicated in the calcium increase after glucose readdiction to starved cells [165].

## 4. Concluding Remarks

It is evident that metabolism has a strong impact on cell signaling and that the interactions between key metabolites and proteins have crucial roles in the cellular response to specific environmental and metabolic conditions. As schematically represented in Figure 6, several metabolites of the central metabolism directly participate in the regulation of different signaling pathways. Metabolites of the glycolysis modulate the SNF1/AMPK/SnRK1 pathway, as well as the PKA pathway and the calcium release mechanism. Molecules belonging to the TCA cycle regulate the retrograde response in yeast and the Ras/PKA pathway. Finally, key amino acids control the TORC1 pathway, through multiple mechanisms.

However, all the mechanisms reviewed here have been discovered in hypothesis-driven studies and large-scale analyses exploring protein-metabolite interactions, as well as their structural and physiological consequences on regulatory proteins, are still needed. One obvious problem is that metabolites mostly interact with metabolic enzymes, while only a small fraction is involved in interactions with signaling proteins to perform regulatory functions [170]. In addition, PMI investigation is hindered by technical problems, since different approaches are needed to study molecules with different chemical properties [7]. Several methodologies for the analysis of the interaction between proteins and small molecule have been developed in recent years (see [7] for a detailed description); however, their application has mainly been focused on the interaction between metabolites and metabolic enzymes. The dynamical nature of PMIs, the high number of interactions involving metabolic enzymes, and also the necessity of validating the binding in vitro and its effect in vivo exacerbate, if possible, the difficulties of the study of PMIs at large scale when focused on regulatory proteins.

Nevertheless, the information acquired from such studies represents actual progress for the comprehension of biological systems. For instance, a systematic analysis of in vivo PMIs in *S. cerevisiae* directed against non-polar small molecules revealed that 20% of protein kinases interact with hydrophobic metabolites [171]. Additionally, system level analysis on yeast cells shifted from glucose to ethanol revealed the requirement of the 14-3-3 protein Bmh1 for yeast growth on ethanol [148]. These examples suggest that our comprehension of the crosstalk between metabolism and cell signaling is still poor, and that the allosteric regulation of regulatory proteins by PMIs could be a common mechanism by which cell signaling could be dynamically connected to metabolism.

## Figures and Tables

**Figure 1 biomolecules-10-00862-f001:**
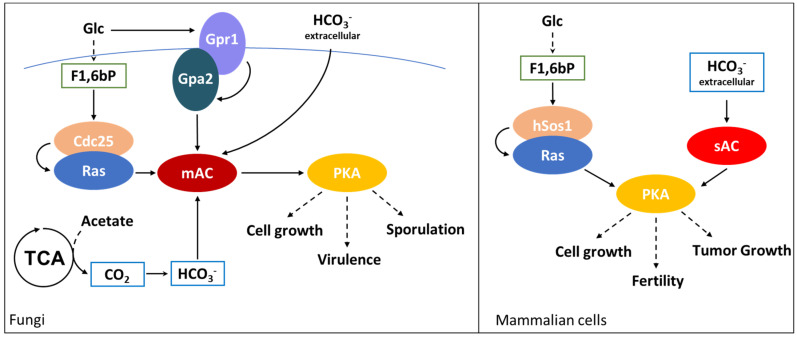
Ras/PKA modulation by PMIs. F1,6bP (green boxes) activates Ras both in yeasts and in mammals, interacting with Cdc25 and hSos1, respectively. Extracellular bicarbonate (blue boxes) interacts with yeast adenylate cyclase, activating virulence promoting processes, such as filamentous growth in *C. albicans* and encapsulation in *C. neoformans*. In mammalian cells, bicarbonate activates soluble adenylate cyclase (sAC) enabling the competence of spermatozoa. Moreover, this interaction promotes cellular growth in oncogenic Ras-driven tumors. In *S. cerevisiae*, acetate oxidation to CO_2_ through the TCA cycle induces sporulation of diploid cells grown in medium containing only this carbon source. Continuous arrows indicate direct activations, dashed lines indicate indirect processes.

**Figure 2 biomolecules-10-00862-f002:**
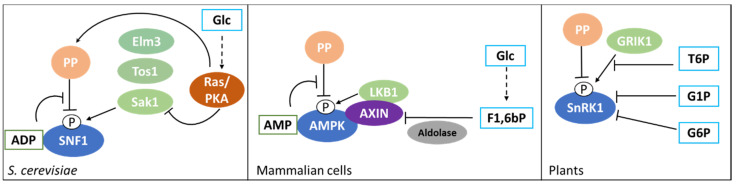
SNF1/AMPK/SnRK1 modulation by nucleotides (green boxes) and glucose metabolites (light blue boxes) in different species. ADP and AMP inhibit dephosphorylation of the regulatory Thr, in yeast and in mammalian cells, respectively. Glucose metabolites inhibit phosphorylation, although the specific mechanism is different in the different species. Additionally, AMPK is also controlled by glycogen through its binding with the Carbon Binding Module (CBS). See text for details. Continuous arrows indicate direct activations, dashed lines indicate indirect processes.

**Figure 3 biomolecules-10-00862-f003:**
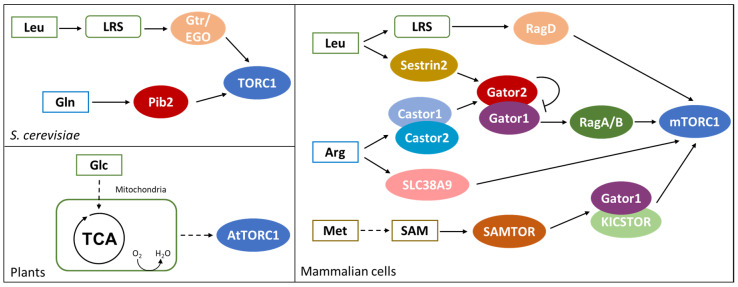
TOR modulation in yeast and in mammalian cells. Both in *S. cerevisiae* and in mammalian cells, leucine activates TOR through Leu-tRNA-synthase (green boxes). In addition, Gln in yeast (blue box), Arg and Met in mammalian cells (purple and orange boxes, respectively) were shown to activate TOR. Contrary to yeast and mammals, in plant cells TOR activity is controlled by glucose oxidation and mitochondrial metabolism. See text for details.

**Figure 4 biomolecules-10-00862-f004:**
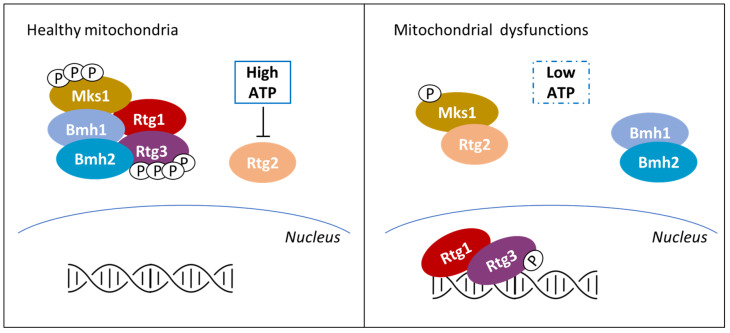
Regulation of the mitochondrial retrograde response in *S. cerevisiae*. In yeast cells presenting healthy mitochondria, Rtg1-Rtg3 complex is recruited to the cytosol by the Mks1-Bmh1/2 complex, with both Rtg3 and Mks1 hyper-phosphorylated. The interaction between Rgt2 and Mks1 is inhibited by ATP interaction with Rtg2 or with other components of the complex. Mitochondrial dysfunction causes a drop in ATP concentration (indicated by the dotted box), leading to Rtg2-Mks1 interaction and hypo-phosphorylated Rtg3 and Mks1. As a consequence, the nuclear localization of the Rtg1-Rtg3 complex induces the expression of RTG genes.

**Figure 5 biomolecules-10-00862-f005:**
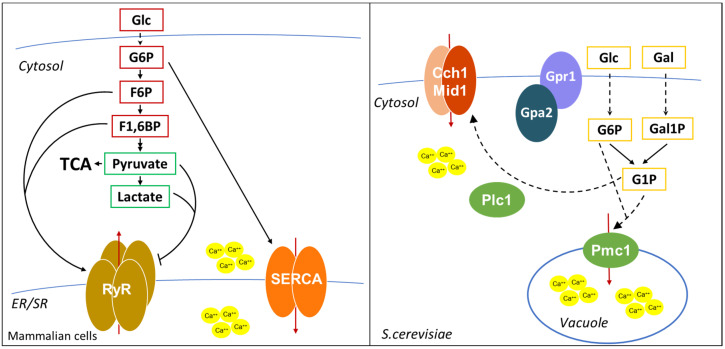
Overview of the effect of glycolytic intermediates on calcium accumulation into the cytoplasm. Metabolites of the upper part of glycolysis (red boxes) are reported to positively influence the activity of RyR channels and SERCA pump, regulating the oscillation of calcium concentration into the cytosol. On the contrary, metabolites of the lower part of glycolysis (green boxes) negatively regulate calcium release from ER/SR through RyR channels. In *S**. cerevisiae* (dotted box) calcium import by the plasma membrane calcium channel is regulated by glucose and galactose availability through their conversion into G6P and G1P.

**Figure 6 biomolecules-10-00862-f006:**
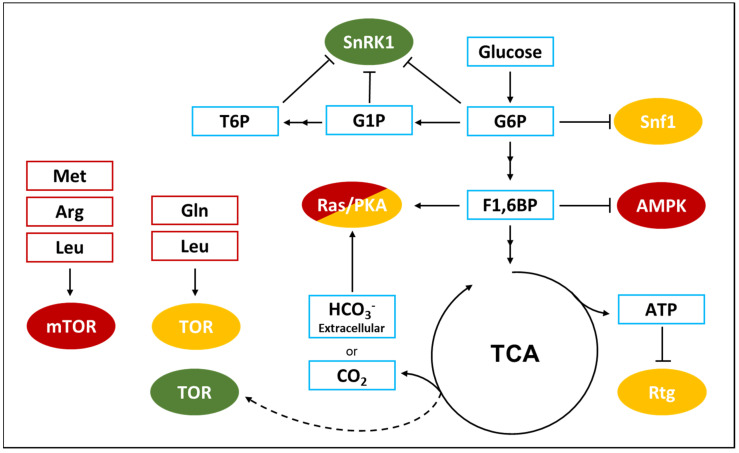
Overview of the Protein-metabolite interactions connecting metabolism and signal transduction pathways in different species. The central carbon metabolism influences the activity of Ras/PKA and SNF1/AMPK/SnRK1 pathways, but also the mitochondrial retrograde response, in yeast, mammals and plants (shown respectively in yellow, red and green). The availability of specific amino acids controls the activity of TOR pathway both in yeast and in mammals, while in plants its activity is influenced by mitochondrial respirations.

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
