# Peer review of "The Regulatory Role of Key Metabolites in the Control of Cell Signaling"

_biomolecules, 2020, doi:10.3390/biom10060862_

Round 1
Reviewer 1 Report
The authors tried to summarize the metabolic products-mediated energy sensing and signaling regulation. The authors have added systematic information in the fields of energy-sensing signaling pathways together with some of the proteins that are well-known enzymes to catalyze metabolic pathways including carbohydrates such as glucose, amino acids, and lipids. The authors entitled the article “protein-metabolite interactions: how metabolism controls cell signaling”. Based on the title of the manuscript, the reviewer considered that the manuscript should mainly describe the conformational change of metabolites and target proteins in structural levels. However, the content of the manuscript has mainly described the signaling induced by the extrinsic and intrinsic change of some of the glucose metabolites such as F1,6BP, phospholipid metabolites such as IP3 and DAG, and amino acid metabolites such as Gln. Thus, the reviewer considered that the manuscript title should change appropriately.
Some of the descriptions provided by the authors are too foggy to understand. For example, on page 2 of section 2 (PMI), the authors’ description as “A clear example of conformational selection ……. [20].” Is very difficult to understand without detailed information for conformational change with binding force and binding chemical groups (moieties). Moreover, it is not clear how glyceraldehyde 3-phosphate dehydrogenase with NAD+ interaction can be compared to T7 polymerase’s conformational change. The reviewer found this kind of issue in several parts of the manuscript.
In the first paragraph in the 3.1 section, it is not clear whether Gpr1/2 can directly interact with Ras or not. Moreover, it is not also clear whether glucose can directly bind with Ras or not. Glucose perception only happens through Hexokinase and GLUT.
The content of the manuscript is very wide. Thus, the reviewer suggests to the authors to focus some of the specific target protein for metabolite-enzyme complex-mediated energy sensing and signaling control with molecular detail mechanisms.
Author Response
Reviewer #1
The authors tried to summarize the metabolic products-mediated energy sensing and signaling regulation. The authors have added systematic information in the fields of energy-sensing signaling pathways together with some of the proteins that are well-known enzymes to catalyze metabolic pathways including carbohydrates such as glucose, amino acids, and lipids. The authors entitled the article “protein-metabolite interactions: how metabolism controls cell signaling”. Based on the title of the manuscript, the reviewer considered that the manuscript should mainly describe the conformational change of metabolites and target proteins in structural levels. However, the content of the manuscript has mainly described the signaling induced by the extrinsic and intrinsic change of some of the glucose metabolites such as F1,6BP, phospholipid metabolites such as IP3 and DAG, and amino acid metabolites such as Gln. Thus, the reviewer considered that the manuscript title should change appropriately.
As suggested by the Reviewer, the title of the manuscript has been changed as follows: “The regulatory role of key metabolites in the control of cell signaling”.
Some of the descriptions provided by the authors are too foggy to understand. For example, on page 2 of section 2 (PMI), the authors’ description as “A clear example of conformational selection ……. [20].” Is very difficult to understand without detailed information for conformational change with binding force and binding chemical groups (moieties). Moreover, it is not clear how glyceraldehyde 3-phosphate dehydrogenase with NAD+ interaction can be compared to T7 polymerase’s conformational change. The reviewer found this kind of issue in several parts of the manuscript.
The aim of citing glyceraldehyde 3-phosphate dehydrogenase and T7 polymerase together was only to bring examples of the different mechanism by which the interaction with a small molecule can affect protein structure. NAD+ controls the activity of glyceraldehyde 3-phosphate dehydrogenase through conformational selection, while T7 DNA polymerase activity is controlled by pairing of the right nucleotide through an induced fit mechanism. However, these examples were not so relevant in this context, so we decided to remove that paragraph (page 2).
In addition, we applied the following revisions:
-page 2 lines 67-69, the sentence has been changed: “NMR and single-molecule spectroscopy studies revealed that proteins can undergo conformational changes both when bound and unbound to a small molecule, due to thermal activated exchange between higher and the lowest energy conformations”.
-page 2 lines 71-73, the sentence has been edited: “Because of this, an allosteric regulation due to the binding of a small molecule on its target can be explained by two different models: conformational selection and induced fit”.
-page 2 lines 76-88, the sentences have been edited: “In this mechanism, the ligand binding takes place after the conformational change and results in the stabilization of the excited conformation, therefore leading to the enrichment of the bound molecular species [18, 19]. On the contrary, in the induced fit model, ligand binding takes place with the basal conformation of the target, before the conformational change, resulting in a transient bound complex still in the initial conformation and characterized by a higher energy. The relaxation of this structure is achieved through the fitting of the target to the bound molecule, as recalled by the definition of this mechanism [16, 20]”.
In the first paragraph in the 3.1 section, it is not clear whether Gpr1/2 can directly interact with Ras or not. Moreover, it is not also clear whether glucose can directly bind with Ras or not. Glucose perception only happens through Hexokinase and GLUT.
Glucose perception in yeast cells is achieved through two different systems. The extracellular system is mediated by the Gpr1/Gpa2 system, while the intracellular one is mediated by Ras. Both Gpa2 and Ras are GTP binding proteins that, in their GTP-loaded form, interact and activate Cyr1. Gpr1 perceives external glucose and promotes Gpa2 activation, while internal glucose is sensed through its conversion into F1,6bP and through F1,6bP interaction with Cdc25, the GEF (Guanine Nucleotide Exchange Factor) activating Ras.
The authors agree with the reviewer that the description of this concept should be reinforced, therefore the following revisions have been applied to revised version of the manuscript:
-page3 lines 124-126, the sentence has been edited: “The extracellular system relies on the G-protein coupled receptor Gpr1, that once bound to glucose, interacts with Gpa2, the α-subunit of a heterotrimeric G-protein complex, promoting its loading with GTP and its release from the complex”.
-page3 lines 130-132 the sentence has been edited: “The two different mechanisms for Ras/PKA activation converge on the adenylate cyclase (AC) Cyr1, whose catalytic activity is triggered by the interaction with Ras2 and Gpa2, resulting in a strong increase of intracellular cAMP concentration”.
-page3 lines 163-166, a new sentence has been added: “This interaction has been proposed to take place at the interface of the complex formed by Cdc25 and Ras; indeed, point mutations of Cdc25 residues involved in the binding impair F1,6bP activation of Ras [9].”
The content of the manuscript is very wide. Thus, the reviewer suggests to the authors to focus some of the specific target protein for metabolite-enzyme complex-mediated energy sensing and signaling control with molecular detail mechanisms.
The authors believe that the aim of the manuscript, in order to be valuable for the scientific community, is to comprehend as much as possible interactions between metabolites and proteins important for the control of signaling pathways. We want to highlight that PMIs can be considered a relevant, yet underestimated, mechanism by which signaling and metabolism can integrate and coordinate one with the other, in addition to the signaling classical control on metabolism through gene expression and post-translational modifications. Different PMIs have been described in the present manuscript with different levels of detail, in light of the fact that the available information is characterized by different degrees of resolution.
Given the aim of this manuscript, the authors chose to describe also less detailed PMIs, such as those involved in mitochondrial retrograde response and calcium signaling, in order to underscore that the described phenomenon do not interest only signaling pathways directly related to metabolism. We hope and believe that our manuscript may prompt other scientists to address this issue, with the final goal to acquire more knowledge and instruments to modulate cell signaling and metabolism in different physio-pathological conditions.
This point has been stressed at page 3 lines 116-119 of the revised manuscript.
Reviewer 2 Report
The manuscript entitled “Protein-Metabolite Interactions: How Metabolism Controls Cell Signaling” by Milanesi R, Coccetti P and Tripodi F represents comprehensive review of the hot topics in cell signalling, how cells have potential to respond to specific environmental and metabolic conditions. It could be great basis also for readers who study pathological changes induced by different environmental and metabolic disruptors.
Check references style: 19 and 31.
Author Response
Reviewer #2
The manuscript entitled “Protein-Metabolite Interactions: How Metabolism Controls Cell Signaling” by Milanesi R, Coccetti P and Tripodi F represents comprehensive review of the hot topics in cell signalling, how cells have potential to respond to specific environmental and metabolic conditions. It could be great basis also for readers who study pathological changes induced by different environmental and metabolic disruptors.
Check references style: 19 and 31.
We really thank the reviewer for his/her positive comments.
The style of the indicated references has been properly adjusted.
Reviewer 3 Report
This review paper by Milanese et al. presents valuable insight and information for the broader scientific community. It introduces and discusses a very interesting and important topic. The regulatory role of metabolism has long been and still is underappreciated. While the goal and the angle of the review are clear, the rationale for the focus on some of the pathways (especially mitochondrial retrograde and calcium signaling) not always is. Sometimes very (too) detailed information on specific mechanisms takes away attention from the main focus and message, while other (equally interesting) mechanisms are discussed much more briefly or not at all. It is obviously not possible to be exhaustive as this is a broad topic (and probably also not the goal and strength of this type of manuscript), but I think some additional pathways and mechanisms should be included for the review to be more balanced. The text is excellent in some places, but needs language editing in others.
Some more specific suggestions:
- The abstract could be phrased more carefully, emphasizing that the focus is on primary metabolism regulating pathways through regulatory metabolites.
- p 2, line 3: I do not understand ‘a similar conceptualization’
- p 3: ‘metabolites that act as second messengers.’ I guess the case is made that metabolites can act as ‘primary’ messengers. That is also why I believe the part on cADPR (further in the text) might not fit entirely in this review (no direct link with primary metabolism). Oterwise, many other secondary messengers should also be considered.
- p 4 and figure 1: Does yeast have a transmembrane AC? Is it not associated with the membrane through lipid modifications? On figure 1, it would be good to indicate that Gpr1 and Gpa2 sense extracellular glucose. This is a prime example of how a nutrient is sensed as a signal. In that respect, I am a bit surprised that there is no mention of the different nutrient ‘transceptors’ identified especially in yeast and likely occurring in other organisms as well. There are also transporters-turned-into-receptors, such as Snf3 and Rgt2, and moonlighting enzymes such as HXK.
- p 6 and figure 2: When discussing SNF1/AMPK/SnRK1 regulation by nucleotide binding, it would be interesting to mention the nucleotide binding gamma subunit Bateman (CBS) domains. For AMPK, AMP binding was not only shown to protect the T-loop form dephosphorylation, it also stimulates phosphorylation and allosterically activates the kinase. For plants, AMP was also reported to inhibit dephosphorylation. Importantly, the SNF1/AMPK/SnRK1 beta subunits also have a carbohydrate binding module, which in AMPK binds glycogen. Its function in SNF1 and SnRK1 is not as clear, although there are (sometimes conflicting) reports of starch and maltose binding. The plant SnRK1 gamma subunit also has acquired such a CBM (producing a hybrid betagamma protein).
- p 7. Note that in plants trehalose does not act as a storage carbohydrate.
- p 8. The TOR pathway in plants is not mentioned, while a lot of progress has been made there in recent years. Interestingly, it was also found to be activated by glucose.
- p 11. This retrograde signalling is more indirect. Does ATP depletion by starvation trigger the same effects? Do similar mechanisms exist in other organisms? Maybe a link could be made with nucleotide status sensing and signalling by AMPK/SNF1/SnRK1?
- p 12. Ca release and signalling is involved in many more different processes in many/most organisms, not just in human cells. As mentioned above, I am not sure the role of cADPR fits well in the scope of the review, unless the aims and focus are defined differently.
- p 14. ‘Hypothesis-driven mechanisms’ is not entirely clear to me. The concluding remarks are interesting and important. It would maybe be good to elaborate a little bit more on the challenges of studying metabolism’s regulatory effects and possible approaches.
Author Response
Reviewer #3
This review paper by Milanese et al. presents valuable insight and information for the broader scientific community. It introduces and discusses a very interesting and important topic. The regulatory role of metabolism has long been and still is underappreciated. While the goal and the angle of the review are clear, the rationale for the focus on some of the pathways (especially mitochondrial retrograde and calcium signaling) not always is. Sometimes very (too) detailed information on specific mechanisms takes away attention from the main focus and message, while other (equally interesting) mechanisms are discussed much more briefly or not at all. It is obviously not possible to be exhaustive as this is a broad topic (and probably also not the goal and strength of this type of manuscript), but I think some additional pathways and mechanisms should be included for the review to be more balanced. The text is excellent in some places, but needs language editing in others.
The choice to describe how PMIs affect mitochondrial retrograde response and calcium signaling is meant to exemplify that metabolism influence upon cell signaling is not limited to pathways related to metabolism, such as those of Ras/PKA, Snf1/AMPK/SnRK1, but can be considered a general principle for the modulation of cellular signaling. This point has been cleared out at page 3 lines 116-119.
We also carefully edited the text where it was not clear enough.
Some more specific suggestions:
- The abstract could be phrased more carefully, emphasizing that the focus is on primary metabolism regulating pathways through regulatory metabolites.
The abstract has been edited as suggested by the Reviewer (page 1, lines 9-24 of the revised manuscript).
- p 2, line 3: I do not understand ‘a similar conceptualization’
This concept has been rephrased as suggested, see page 2 lines 50-52.
- p 3: ‘metabolites that act as second messengers.’ I guess the case is made that metabolites can act as ‘primary’ messengers. That is also why I believe the part on cADPR (further in the text) might not fit entirely in this review (no direct link with primary metabolism). Otherwise, many other secondary messengers should also be considered.
We agree with the Reviewer, so we deleted the sentence “that act as second messengers” (page 3, line 141 of the revised manuscript).
We agree with the point on cADPR, so we strongly reduce that part in the manuscript (pages 13-14 of the revised manuscript). We just left a mention to cADPR regulation in the first part of the chapter generally describing calcium signaling (lines 600-604). We also removed cADPR from Figure 5.
- p 4 and figure 1: Does yeast have a transmembrane AC? Is it not associated with the membrane through lipid modifications? On figure 1, it would be good to indicate that Gpr1 and Gpa2 sense extracellular glucose. This is a prime example of how a nutrient is sensed as a signal. In that respect, I am a bit surprised that there is no mention of the different nutrient ‘transceptors’ identified especially in yeast and likely occurring in other organisms as well. There are also transporters-turned-into-receptors, such as Snf3 and Rgt2, and moonlighting enzymes such as HXK.
As correctly indicated by the Reviewer, yeast AC is not a transmembrane protein, but it is just associated to the membranes (Belotti et al., Biochim Biophys Acta. 2012 Jul;1823(7):1208-16. doi: 10.1016/j.bbamcr.2012.04.016), so we changed in the Figure 1 “tmAC” with “mAC” (membrane AC). We also added Gpr1 and Gpa2 to Figure 1, as suggested. However, we point out that the focus of this manuscript is to highlight the role of primary intracellular metabolism in regulating the activity of signal transduction pathways. The perception of nutrients is a broad topic and the author intentionally choose not to describe mechanisms of perception of nutrients at the cellular surface, which are excellently reviewed elsewhere. The focus of the manuscript is closely related to that, but its slightly different. The authors want to elucidate the control of metabolized nutrients on signaling.
Taking into consideration the example of glucose perception, Hyun Youk and Alexander van Oudenaarden in 2009 clearly demonstrated that the perception of glucose availability through Snf3 and Rgt2 set the expression of specific glucose transporters and therefore specific glucose transport rate, metabolism and growth rate. In this manuscript we highlight that glucose metabolites can influence, on the way back, cell signaling. For example, glucose metabolized into F1,6bP, can influence the activity of Ras/PKA and AMPK in yeast and mammals respectively.
However, the authors agree with the Reviewer that, for comprehensiveness, it is important to underscore that a relevant part of the crosstalk between metabolism and signaling begins with nutrient perception on the plasma membrane. This point has been cleared at page 3 lines 107-115. Figure 1 has been corrected accordingly.
- p 6 and figure 2: When discussing SNF1/AMPK/SnRK1 regulation by nucleotide binding, it would be interesting to mention the nucleotide binding gamma subunit Bateman (CBS) domains. For AMPK, AMP binding was not only shown to protect the T-loop form dephosphorylation, it also stimulates phosphorylation and allosterically activates the kinase. For plants, AMP was also reported to inhibit dephosphorylation. Importantly, the SNF1/AMPK/SnRK1 beta subunits also have a carbohydrate binding module, which in AMPK binds glycogen. Its function in SNF1 and SnRK1 is not as clear, although there are (sometimes conflicting) reports of starch and maltose binding. The plant SnRK1 gamma subunit also has acquired such a CBM (producing a hybrid betagamma protein).
As suggested by the Reviewer, we added information about the role of the Bateman (CBS) domain in the binding of AMP with AMPK and in the regulation of its catalytic activity. See page 7 lines 292-296. Bateman domains has been mentioned also in the discussion of ADP protection of Snf1. See page 6 line 267-271.
However, the authors cannot find in the literature any evidence of AMP protecting SnRK1 from dephosphorylation. On the contrary, different papers show that AMP do not have any effect on the activity of SnRK1; moreover Bateman (CBS) domains in the hybrid βγ subunit have been reported to be mutated in the residues involved in the binding of adenosyl-nucleotides (see Emanuelle et al., the Plant Journal 2015, doi: 10.1111/tpj.12813. Zhao 2019, Protein Expression and Purification doi.org/10.1016/j.pep.2019.05.007 and Emanuelle et al., 2016 Trends in Plant Science doi.org/10.1016/j.tplants.2015.11.001). This point has been reported at page 8 lines 352-355.
Finally, as suggested by the Reviewer, a paragraph describing the role of AMPK CBM in the control of the kinase has been added. The involvement of CBM in the control of Snf1 and SnRK1 has been briefly stated in that context. See page 7 and 8 lines 325-336.
- p 7. Note that in plants trehalose does not act as a storage carbohydrate.
Page 8 line 358-360, the sentence on trehalose has been edited, as suggested.
- p 8. The TOR pathway in plants is not mentioned, while a lot of progress has been made there in recent years. Interestingly, it was also found to be activated by glucose.
We thank the Reviewer for her/his useful suggestion. We added a paragraph on TOR signaling in plants (pages 11-12, lines 509-539 of the revised manuscript). Indeed, glucose activation on TOR is very interesting, although the mechanism of this activation is still largely unknown.
- p 11. This retrograde signalling is more indirect. Does ATP depletion by starvation trigger the same effects? Do similar mechanisms exist in other organisms? Maybe a link could be made with nucleotide status sensing and signalling by AMPK/SNF1/SnRK1?
Retrograde signaling in yeast is activated by a nutritional shift from glucose to non-fermentable carbon sources, for example acetate (as added in the revised manuscript, page 12, lines 544-547). The adaptation to the growth in this condition requires the expression of several genes under the control of the RTG (first of all of the citrate synthase CIT2, the main marker for RTG activation in yeast). This could be due to a transient drop in ATP concentration, but this has still to be demonstrated. Meanwhile, in the literature there is no evidence of the involvement of Snf1/AMPK in the control of RTG; on the contrary, it is more likely that Snf1 and RTG pathways independently respond to ATP depletion. This point has been cleared out at page 12 lines 577-579.
The retrograde signaling of mitochondrial dysfunction to nucleus is an important process also for the homeostasis of mammalian and plants cells. However, the triggering of this signaling pathway mainly depends on the alteration of mitochondrial membrane potential, followed by a strong increase in cytoplasmic calcium. Therefore, it is based on an electrochemical signal and do not involve a protein metabolite interaction. We mentioned these concepts at page 12 at lines 549-552.
Nevertheless, ATP control of Rtg2-Mks1 interaction have been described also in Kluveromyces lactis and Kluveromyces waltii clearly suggesting the conservation of this mechanism among Fungi.
- p 12. Ca release and signalling is involved in many more different processes in many/most organisms, not just in human cells. As mentioned above, I am not sure the role of cADPR fits well in the scope of the review, unless the aims and focus are defined differently.
As suggested by the Reviewer, we added a paragraph (page 14, lines 619-648) on the role of calcium and its regulation in Saccharomyces cerevisiae. Although no molecular details are available, it is an interesting and relevant part, which needed to be added to the manuscript. Instead, we chose not to add a paragraph on the role of calcium in plant cells, which is however very important as a regulator of different processes (hormone responses, development, plant-pathogen interaction, symbiosis, salt stress and circadian rhythm), because we couldn’t find any information on PMIs regulating calcium channels in plants.
In addition, following the Reviewer’s suggestion, we strongly reduced the part on cADPR (page 14, lines 600-604).
- p 14. ‘Hypothesis-driven mechanisms’ is not entirely clear to me. The concluding remarks are interesting and important. It would maybe be good to elaborate a little bit more on the challenges of studying metabolism’s regulatory effects and possible approaches.
As suggested by the Reviewer, the concluding remarks have been edited (see page 16). Some of the technical problems in analysing regulatory PMIs have been outlined, moreover the importance of such mechanisms in the control of biological systems has been stated more clearly. We rephrased the concept of “hypothesis driven mechanism” (page 16 line 665-667), stressing the fact that all the PMIs reviewed in the manuscript have been discovered addressing the specific hypothesis that metabolite can alter the activity of proteins. On the contrary, we propose that large scale studies focused on the identification of PMIs involving regulatory proteins would add useful knowledge in the field.
Round 2
Reviewer 1 Report
There are no more concerns.